# Comparative Study of Muscle Hardness during Water-Walking and Land-Walking Using Ultrasound Real-Time Tissue Elastography in Healthy Young People

**DOI:** 10.3390/jcm12041660

**Published:** 2023-02-19

**Authors:** Naoya Tanabe, Yasuko Nishioka, Kyosuke Imashiro, Hiromi Hashimoto, Hiroki Kimura, Yasuhiro Taniguchi, Koya Nakai, Yasunori Umemoto, Ken Kouda, Fumihiro Tajima, Yasuo Mikami

**Affiliations:** 1Department of Rehabilitation Medicine, Wakayama Medical University, 811-1 Kimiidera, Wakayama 641-8509, Japan; 2Division of Rehabilitation, Wakayama Medical University Hospital, 811-1 Kimiidera, Wakayama 641-8510, Japan; 3Department of Rehabilitation Medicine, Graduate School of Medical Science, Kyoto Prefectural University of Medicine, 465 Kawaramachi-Hirokoji, Kamigyo-ku, Kyoto 602-8566, Japan

**Keywords:** muscle hardness, ultrasound real-time tissue elastography, aerobic exercise

## Abstract

Compared with land-walking, water-walking is considered to be beneficial as a whole-body exercise because of the characteristics of water (buoyancy, viscosity, hydrostatic pressure, and water temperature). However, there are few reports on the effects of exercise in water on muscles, and there is no standard qualitative assessment method for muscle flexibility. Therefore, we used ultrasound real-time tissue elastography (RTE) to compare muscle hardness after water-walking and land-walking. Participants were 15 healthy young adult males (24.8 ± 2.3 years). The method consisted of land-walking and water-walking for 20 min on separate days. The strain ratio of the rectus femoris (RF) and medial head of gastrocnemius (MHGM) muscles were measured before and immediately after walking using RTE to evaluate muscle hardness. In water-walking, the strain ratio significantly decreased immediately after water-walking, with *p* < 0.01 for RF and *p* < 0.05 for MHGM, indicating a significant decrease in muscle hardness after water-walking. On the other hand, land-walking did not produce significant differences in RF and MHGM. Muscle hardness after aerobic exercise, as assessed by RTE, was not changed by land walking but was significantly decreased by water walking. The decrease in muscle hardness induced by water-walking was thought to be caused by the edema reduction effect produced by buoyancy and hydrostatic pressure.

## 1. Introduction

Health-related fitness consists of five factors: cardiorespiratory endurance, muscular strength, muscular endurance, body composition, and flexibility [1]. Sit-and-reach test performance [2] and finger floor distance (FFD) [3] are used as measurements for evaluating body flexibility. However, these measures are complex quantitative flexibility assessments of the muscles and joints, and it is difficult to qualitatively evaluate the flexibility of individual muscles. Ultrasound real-time tissue elastography (RTE) was recently developed to evaluate tissue stiffness in real-time images, and high intra-examiner reproducibility was reported [4]. Ultrasound elastography methods can be broadly classified into two types: shear wave elastography (SWE), which measures the conduction velocity of shear waves, and strain elastography (SE), which measures the strain by deforming tissue through the application of external stress. RTE, a type of SE, is a real-time imaging technique that converts the micro-displacement generated by the examiner’s application of minute pressure through a probe into the strain. RTE can assess the distribution of stiffness within a particular muscle. In humans, light pressure on the epidermis with a hand-held ultrasound transducer produces strain within the subcutaneous tissue. Stiff tissue has a lower strain rate than soft tissue, and RTE visualizes the strain distribution within the tissue as a translucent, color-coded tissue elasticity map which is superimposed on the B-mode image. By comparing this strain rate to that of a reference material, the hardness of a particular tissue can be evaluated semi-quantitatively (strain ratio) [5,6,7]. This method has been used to differentiate benign and malignant lesions of the mammary gland [8], lymph nodes [9], and liver [10]. In recent years, this approach has been increasingly used to evaluate the muscle hardness of skeletal muscle [11,12,13]. Although some studies have reported increased muscle hardness after strength training using RTE [6,7,14], to the best of our knowledge, few studies have examined changes in lower extremity muscle hardness after aerobic exercise. Among aerobic exercises, water exercise has been reported to improve aerobic capacity [15], the range of joint motion [16], pain [17], and FFD [18] because of the physical properties of water (density, buoyancy, viscosity, and the effects of hydrostatic pressure and water temperature [19,20] compared with land-walking. However, it is unclear whether water exercise reduces muscle stiffness. Consequently, the purpose of this study was to clarify the changes in muscle hardness of the rectus femoris and the medial head of the gastrocnemius muscle after land-walking and water-walking using RTE. The results were also compared with the FFD, which has traditionally been used to assess flexibility.

## 2. Materials and Methods

### 2.1. Participants

The participants were 15 healthy young adult male volunteers, as shown in Table 1. All muscle hardness measurements were performed on the right lower extremity. Participants had no history of orthopedic disease in the lower extremity.

### 2.2. Walking Program

The walking time for land-walking and water-walking was 20 min. Exercise loading was performed using the category ratio scale (CR-10) [21] recommended by the American College of Sports Medicine (ACSM) [22] in the range of 3 (moderate) to 4 (somewhat strong). For land-walking, participants were instructed to walk back and forth along a 30 m flat corridor (temperature controlled to 28 °C by air conditioning). Water-walking was performed in a flat heated pool (water temperature 33 °C). A temperature of 33 °C is considered thermoneutral and is recommended for in-water exercise [23]. The length and width of the pool were 4.7 m and 2.9 m, respectively. The water level was 1 m, which was approximately equal to the height of each participant’s umbilical region. The umbilical level for immersion reduces by half the weight and induces fewer alterations in the posture and less turbulence than a deeper immersion [24].

### 2.3. Assessment of Muscle Hardness

RTE measurements were performed with Logiq^®^ S8 ultrasound (GE Healthcare, Milwaukee, WI, USA) using an 8–12 MHz multi-frequency linear probe (11L-D; GE Healthcare, Milwaukee, WI, USA). The measured muscles were the rectus femoris (RF) and the medial head of the gastrocnemius (MHGM) muscle, which are the muscles primarily targeted when evaluating muscle activity during walking and have been reported in the past as measurement sites for RTE. The measurement limb and probe position were as follows: the rectus femoris muscle was placed in the supine position at the midpoint between the superior anterior iliac spine and the proximal end of the patella [25,26], and the medial head of the gastrocnemius muscle was placed in the prone position with the knee extended at the level where the muscle thickness of the medial head of the gastrocnemius muscle was at its maximum in the mediolateral direction [4] (Figure 1). The leg to be measured was unified on the right side.

Line markers were drawn on the skin surface at the location to be scanned for both muscles so that measurements could be taken at the same location before and after walking. Measurements were scanned by manually applying a light repetitive compression (rhythmical compression relaxation cycle) to the transducer at the probe position. A 3 mm thick gel pad (Echo Gel PAD, Yasojima Proceed Co., Ltd., Hyogo, Japan) was placed between the transducer and the skin. The scanned images were placed in four regions of interest (ROI), 5 mm in the medial head of the rectus femoris and gastrocnemius muscles and 3 mm in the gel pad, respectively. The strain rate in each ROI was automatically measured by built-in software (version R4.2.53), and the strain ratio (reference ratio/muscle; ROI of the reference divided by ROI of the muscle) was calculated. The values of the strain ratio were the average of the four locations (Figure 2).

### 2.4. Finger Floor Distance (FFD)

The measurement of the FFD [3] was performed using a 40 cm high measuring table. The starting position was a stationary standing position on the measurement table (step width 5 cm), and participants were instructed to flex their trunk forward and lower their upper limbs to the left and right at their own timing. Using a tape measure, the position of the midpoint at the line connecting the tips of the right and left middle fingers was measured.

Values were measured in 0.1 cm increments with the height of the measuring table set at 0 cm, with (−) indicating when the fingers stopped at a point higher than the surface of the table and (+) indicating when they passed the surface of the table. Measurements were taken only once, and if flexion of the knee joint was observed, the measurement was stopped.

### 2.5. Statistical Analysis

Measurement results are presented as the mean ± standard deviation. The Shapiro–Wilk test was used to determine the normality of the measured values. Comparisons of strain ratio and FFD before and after each walk, as well as the change in strain ratio (Δ) before and after land-walking and water-walking, were made using the corresponding *t*-tests. All statistical analyses were performed using GraphPad Prism 7 (GraphPad Software Inc., La Jolla, CA, USA) with a significance level of less than 5%.

## 3. Results

The strain ratio for land-walking did not significantly change in RF (2.6 ± 1.3 before walking, 2.6 ± 1.2 after walking, *p* = 0.84, 95% CI: −0.3562 to 0.4295) and MHGM (3.5 ± 1.4 before walking, 3.6 ± 1.6 after walking, *p* = 0.41, 95% CI: −0.2041 to 0.4681). No significant changes were observed in either case (Figure 3A). On the other hand, water-walking showed a significant decrease in both RF (3.4 ± 1.0 before walking, 2.2 ± 0.6 after walking, *p* < 0.0001, 95% CI: −1.576 to −0.7142) and MHGM (3.9 ± 1.7 before walking, 2.8 ± 1.0 after walking, *p* = 0.0143, 95% CI: −1.968 to −0. 2591), with a significant decrease in both muscles (Figure 3B).

Comparing the change in the strain ratio before and after land-walking and water-walking, both ΔRF (land-walking 0.0 ± 0.7 vs. water-walking −1.1 ± 0.8, *p* = 0.0006, 95% CI: −1.752 to −0.6121) and ΔMHGM (land-walking 0.1 ± 0.6 vs. water-walking −1.1 ± 1.5, *p* = 0.0277, 95% CI: −2.333 to −0.1577) were significantly lower after water-walking (Figure 4).

Regarding FFD, the change before and after walking (land-walking 0.4 ± 1.5 cm vs. water-walking 3.0 ± 2.7 cm, *p* < 0.01, 95% CI: 1.006 to 4.327) was also significantly longer after water-walking.

## 4. Discussion

The current study used RTE in healthy young participants to examine the changes in muscle hardness resulting from land-walking and water-walking. In this study, RF and MHGM strain ratios were measured before and after walking, and only water walking showed a significant decrease in the strain ratio immediately after walking. The change in muscle hardness was significantly greater for water walking than for land walking.

Previous studies of muscle hardness after strength training have shown that muscle hardness increases after exercise [7]. These phenomena were thought to be caused by the increased intramuscular fluid content caused by repetitive muscle contractions [27,28], the increased intramuscular blood flow caused by increased capillary pressure and permeability, altered vascular, and extravascular osmotic gradients, which were caused by metabolite deposition in the activated muscle, or a combination of these factors [29,30,31]. The reason muscle hardness did not change after land-walking is likely due to the fact that the quadriceps muscle during walking was under less load on the muscle, as the muscle activity to maximum voluntary contraction (%MVC) was 2.3% [32] in healthy young subjects. The increase in blood flow to the lower leg caused by aerobic exercise activates the muscle venous pump and suppresses the occurrence of leg edema [33], possibly preventing muscle hardness from increasing easily.

The decrease in muscle hardness immediately after water-walking may be related to the buoyancy and hydrostatic pressure of the water. Buoyancy may cause lower muscle activity in the lower extremities because of the unloading effect of body weight. Comparative studies examining %MVC in the water and while walking on land have also reported that %MVC was significantly lower in water [34]. The effects of hydrostatic pressure have been reported in previous studies as follows. After 30 min of standing still in a 110-cm-deep pool, lower limb volume was reported to be significantly reduced by the compression of the lower limb by water hydrostatic pressure (wHP) [35]. Stevin’s law [36] states that at a depth of 110 cm, a wHP of 81 mmHg was applied at the bottom of the pool, decreasing by 0.73 mmHg for each 1 cm rise. Thus, in the standing position, a wHP of approximately 60 mmHg was estimated to be applied at the center of the perineum. This effect is thought to be caused by the reduction in leg edema caused by the interstitial fluid being pushed into the lymphatic vessels [37] during exercise under lower limb pressure because of strong hydrostatic pressure. This effect is similar to the findings of a previous study which reported that a decrease in volume was caused by a decrease in extracellular fluid in the lower extremity after walking while wearing graduated compression stockings (GCS) [38]. The leg volume decreased by 192 ± 20 mL during the 3 h head-out water immersion, which accounted for 3.5% of the average leg volume of the pre-immersion period [39]. One limitation of the current study is that participants were healthy young men, so it is possible that muscle fatigue did not occur after 20 min of walking. Future studies should include older people and patients with osteoarticular disease who have limited mobility under a load. In addition, it is unclear how long the decrease in muscle hardness after water-walking persisted, and follow-up measurements over time are necessary. Furthermore, the exercise load index during walking was measured by subjective exercise intensity in the current study. It will be necessary for future studies to set an objective walking speed, as in studies using treadmills and underwater treadmills. As a future prospect for this study, water exercise has been reported to be effective for various diseases, such as musculoskeletal diseases [40], neuromuscular diseases [41], stroke [42], and cardiac diseases [43], and we intend to study the effects of water exercise on muscle hardness in each disease.

## 5. Conclusions

The muscle hardness of RF and MHGM during aerobic exercise in healthy young subjects was found to decrease after water walking, with no change after land-walking. The decrease in muscle hardness after water-walking may have been induced by a decrease in muscle activity caused by buoyancy and hydrostatic pressure and the effect of a decrease in extracellular fluid.

## Figures and Tables

**Figure 1 jcm-12-01660-f001:**
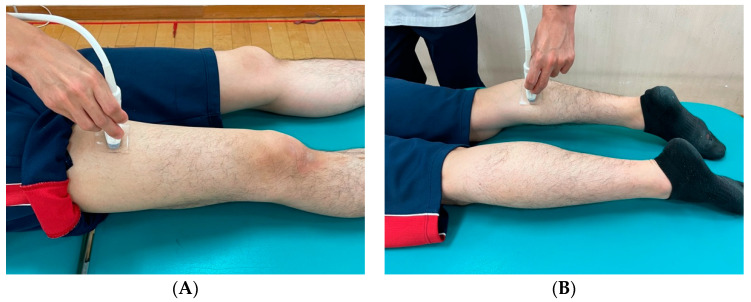
Position of participants during real-time strain elastography. (**A**) Rectus femoris; (**B**) Medial head of gastrocnemius muscle.

**Figure 2 jcm-12-01660-f002:**
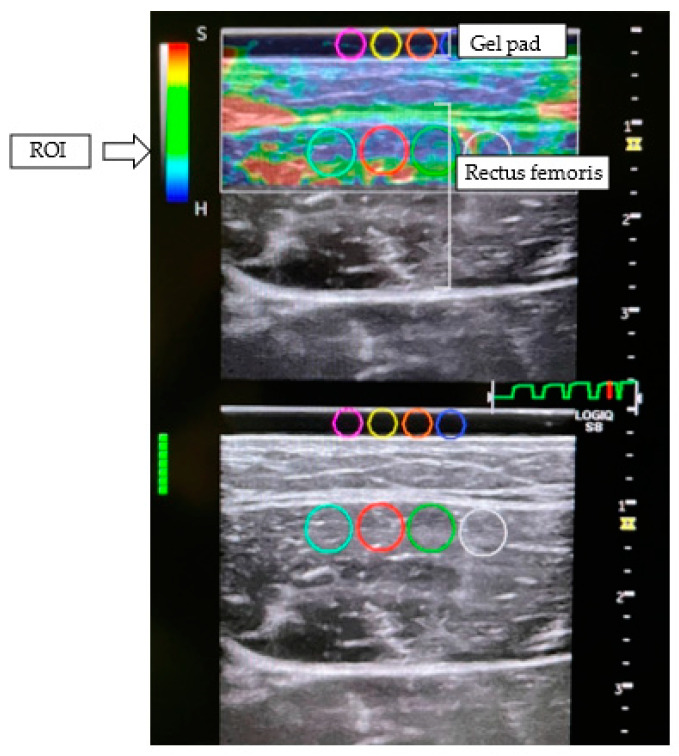
Example of elastography measurement of rectus femoris. Circles on the screen indicate regions of interest (ROI). Four ROI of 5 mm in the rectus femoris muscle and 3 mm in the gel pad were placed at each location.

**Figure 3 jcm-12-01660-f003:**
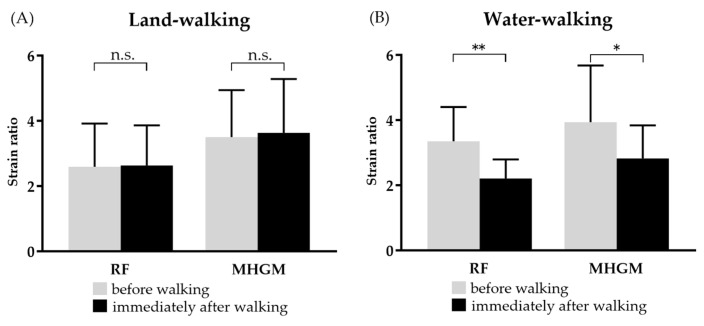
Changes in muscle hardness of RF and MHGM before and after land-walking and water-walking. Land-walking did not change muscle hardness after walking (**A**), but water-walking decreased RF and MHGM muscle hardness after walking (**B**). * *p* < 0.05, ** *p* < 0.01. RF: rectus femoris, MHGM: medial head of gastrocnemius muscle.

**Figure 4 jcm-12-01660-f004:**
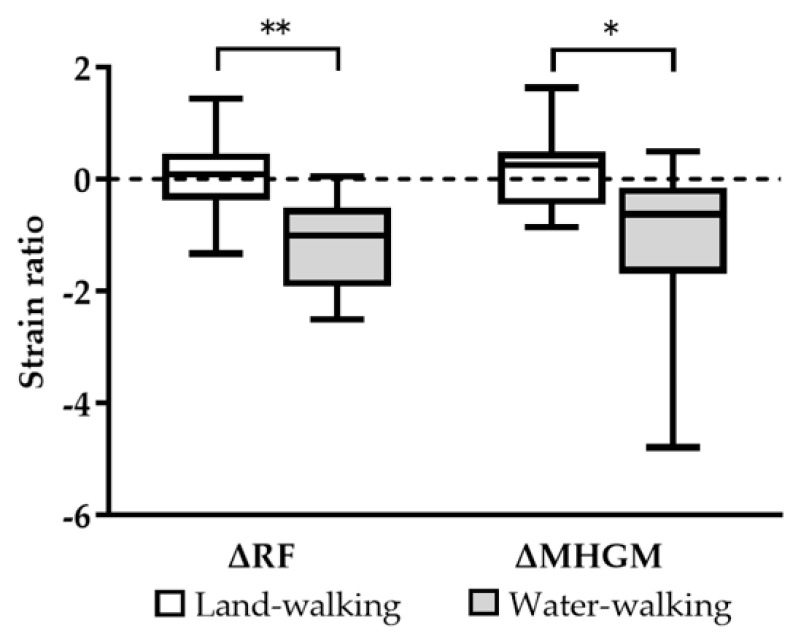
Comparison of the amount of change in muscle hardness (Δ) before and after land-walking and water-walking. Water-walking significantly reduced muscle hardness in RF and MHGM compared to land-walking. * *p* < 0.05, ** *p* < 0.01. RF: rectus femoris, MHGM: medial head of gastrocnemius muscle.

**Table 1 jcm-12-01660-t001:** Anthropometric characteristics of the subjects.

	Healthy Young Adult Male (*n* = 15)
Age (years)	24.8 ± 2.3
Height (cm)	171.5 ± 4.3
Body weight (kg)	64.2 ± 11.4
Body mass index (kg/m^2^)	21.8 ± 3.7

Data are mean ± standard deviation (SD).

## Data Availability

The datasets used and/or analyzed during this study are available from the corresponding author upon reasonable request.

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
