# Peer review of "Comparative Study of Muscle Hardness during Water-Walking and Land-Walking Using Ultrasound Real-Time Tissue Elastography in Healthy Young People"

_jcm, 2023, doi:10.3390/jcm12041660_

Round 1

Reviewer 1 Report

The manuscript is really interesting. The materials and methods are well written and detailed. The results are clearly displayed.

I only suggest extending the discussion section dedicated to clarifying what implications the results of this interesting study may have in the future. For example, reference can be made to the different types of diseases in which exercise in water is used for rehabilitation purposes (for example, I suggest mentioning Coraci, D.; Tognolo, L.; Maccarone, M.C.; Santilli, G.; Ronconi, G.; Masiero, S. Water-Based Rehabilitation in the Elderly: Data Science Approach to Support the Conduction of a Scoping Review. Appl. Sci. 2022, 12, 8999. https://doi.org/10.3390/app12188999; Faíl LB, Marinho DA, Marques EA, Costa MJ, Santos CC, Marques MC, Izquierdo M, Neiva HP. Benefits of aquatic exercise in adults with and without chronic disease-A systematic review with meta-analysis. Scand J Med Sci Sports. 2022 Mar;32(3):465-486. doi: 10.1111/sms.14112. Epub 2021 Dec 24. PMID: 34913530.).

Author Response

Dr. Emmanuel Andrès

Editor-in-Chief

Journal of Clinical Medicine

16 February 2023

Dear Editors:

On behalf of my co-authors, I thank you for the opportunity to revise our manuscript (ID : jcm-2225009), titled “Comparative study of muscle hardness during water-walking and land-walking using ultrasound real-time tissue elastography in healthy young people”. The reviewers’ comments were very insightful and helped us in revising and improving our manuscript.

We have carefully addressed all the reviewers’ comments in our revised manuscript. We hope that our responses and revisions have adequately addressed the reviewers’ concerns and that the revised manuscript will now meet the high standards required for publication in your esteemed journal.

The manuscript has been rechecked and the necessary changes have been made in accordance with the reviewers’ suggestions. The responses to all comments have been prepared and given below. We have applied the “Track Changes” feature of MS Word to highlight the revisions in our updated manuscript.

Thank you for your consideration. We look forward to hearing from you.

Sincerely,

Yasuko Nishioka, MD, PhD

Department of Rehabilitation Medicine, Wakayama Medical University, 811-1 Kimiidera, Wakayama, 640-8509, Japan

Telephone: +81-73-447-0664

Fax: +81-73-446-6475

E-mail: kopandap627@yahoo.co.jp

Response to Reviewer 1 Comments

Point 1:

The manuscript is really interesting. The materials and methods are well written and detailed. The results are clearly displayed.

Response 1:

We would like to thank the Reviewer for their time and effort in reviewing our manuscript and providing comments and suggestions, which have considerably helped us to improve our manuscript. We have answered each of your points below and hope that our responses and revisions address all your comments.

Point 2:

I only suggest extending the discussion section dedicated to clarifying what implications the results of this interesting study may have in the future. For example, reference can be made to the different types of diseases in which exercise in water is used for rehabilitation purposes (for example, I suggest mentioning Coraci, D.; Tognolo, L.; Maccarone, M.C.; Santilli, G.; Ronconi, G.; Masiero, S. Water-Based Rehabilitation in the Elderly: Data Science Approach to Support the Conduction of a Scoping Review. Appl. Sci. 2022, 12, 8999. https://doi.org/10.3390/app12188999; Faíl LB, Marinho DA, Marques EA, Costa MJ, Santos CC, Marques MC, Izquierdo M, Neiva HP. Benefits of aquatic exercise in adults with and without chronic disease-A systematic review with meta-analysis. Scand J Med Sci Sports. 2022 Mar;32(3):465-486. doi: 10.1111/sms.14112. Epub 2021 Dec 24. PMID: 34913530.).

Response 2:

As you indicated, we have expanded the discussion section.

"As a future prospect for this study, water exercise has been reported to be effective for various diseases such as musculoskeletal diseases [42], neuromuscular diseases [43], stroke [44], and cardiac diseases [45], and we intend to study the effects of water exercise on muscle hardness in each disease.”  was added to L.205-208.

Reviewer 2 Report

In general 

The MS is interesting. However, some corrections may improve it.

Introduction

-Please explain more on why walking aerobic exercise may be improve leg ROM and which muscle should be involved.

-Please explain why FFD was selected for flexibility assessment in this study and how it related to RF and MG muscle hardness.

Method

-Was there any temperature control for land-based walking in this experiment? 

-Please explain more why this water level was used in this study.

-L. 86 please check grammatical error.

-L.87 abbre can be used here for muscles

-Please express which leg had been investigated for muscle hardness assessment.

-Please explain or provide reference why ROI locations were analyzed near the superior aponeunosis.

-Please provide the statistic analysis for the comparison between walking conditions.

Results

-Was there any difference between the strain ratio in the pre-muscle state of between walking conditions?

-Please express the data are presented as mean (SD) ...? for Fig. 3 และ 4 median (IQR)?

-L.147-150 if these data were not normal distribution. Median and IQR should be expressed instead of mean (SD)

-L.151-152 please provide the unit

Discussion

-L.172 Is there less load on the muscle also happens for MG m during land-based walking?

-Please provide more the factors that affect the changes in muscle hardness.

Conclusion

-Please consider to use "a decrease extracellular fluid" rather than "edema reduction". or please explain why leg edema increases during walking in that condition.

Author Response

Dr. Emmanuel Andrès

Editor-in-Chief

Journal of Clinical Medicine

16 February 2023

Dear Editors:

On behalf of my co-authors, I thank you for the opportunity to revise our manuscript (ID : jcm-2225009), titled “Comparative study of muscle hardness during water-walking and land-walking using ultrasound real-time tissue elastography in healthy young people”. The reviewers’ comments were very insightful and helped us in revising and improving our manuscript.

We have carefully addressed all the reviewers’ comments in our revised manuscript. We hope that our responses and revisions have adequately addressed the reviewers’ concerns and that the revised manuscript will now meet the high standards required for publication in your esteemed journal.

The manuscript has been rechecked and the necessary changes have been made in accordance with the reviewers’ suggestions. The responses to all comments have been prepared and given below. We have applied the “Track Changes” feature of MS Word to highlight the revisions in our updated manuscript.

Thank you for your consideration. We look forward to hearing from you.

Sincerely,

Yasuko Nishioka, MD, PhD

Department of Rehabilitation Medicine, Wakayama Medical University, 811-1 Kimiidera, Wakayama, 640-8509, Japan

Telephone: +81-73-447-0664

Fax: +81-73-446-6475

E-mail: kopandap627@yahoo.co.jp

Response to Reviewer 2 Comments

Point 1:

In general 

The MS is interesting. However, some corrections may improve it.

Response 1:

We would like to thank the Reviewer for their time and effort in reviewing our manuscript and providing comments and suggestions, which have considerably helped us to improve our manuscript. We have answered each of your points below and hope that our responses and revisions address all your comments.

Point 2:

Introduction

Please explain more on why walking aerobic exercise may be improve leg ROM and which muscle should be involved.

Response 2:

Previous studies using water-walking as a postoperative exercise therapy after TKA surgery have reported improved ROM of the knee joint due to the reduction of edema around the knee. To our knowledge, there are no detailed research reports on which muscles should be used during water-walking.

Point 3:

Please explain why FFD was selected for flexibility assessment in this study and how it related to RF and MG muscle hardness.

Response 3:

FFD was selected because it is a commonly used assessment method for lower extremity flexibility evaluation. Only RF and MHGM have been reported for RTE measurements on lower extremity muscles, and their direct relationship to FFD is unknown, but they were adopted as the measurement sites for this study.

Point 4:

Method

Was there any temperature control for land-based walking in this experiment? 

Response 4:

The land-walk was conducted in a corridor where the temperature was controlled at 28°C by air conditioning.

 "(temperature controlled to 28°C by air conditioning)" was added to L.78.

Point 5:

Please explain more why this water level was used in this study.

Response 5:

"The umbilical level for immersion reduces by half the weight and induces fewer alterations of the posture and less turbulence than a deeper immersion [24]. " was added to L.82-84.

Point 6:

  1. 86 please check grammatical error

Response 6:

Fixed a grammatical error in L.86.

Revised to "The measured muscles were the rectus femoris" and was listed in L.88.

Point 7:

L.87 abbre can be used here for muscles

Response 7:

Abbreviations for RF and MHGM were added to L.88-89.

Point 8:

Please express which leg had been investigated for muscle hardness assessment.

Response 8:

"The leg to be measured was unified on the right side." was added to L.96-97.

Point 9:

Please explain or provide reference why ROI locations were analyzed near the superior aponeunosis.

Response 9:

To the best of our knowledge, there are no reports defining the position of the ROI setting during RTE measurement.

In this study, the ROI was established using near the superior aponeurosis, which is easy to determine, as the index to reduce the variability of measurements.

Point 10:

Please provide the statistic analysis for the comparison between walking conditions.

Response 10:

This study used only walking time and subjective exercise intensity (CR-10) to set exercise intensity, and did not measure measures such as walking speed or walking distance for each subject.

Point 11:

Results

Was there any difference between the strain ratio in the pre-muscle state of between walking conditions?

Response 11:

There were no significant differences in muscle hardness between land-walking and water-walking conditions for RF and MHGM before walking.

Point 12:

Please express the data are presented as mean (SD) ...? for Fig. 3 และ 4 median (IQR)?

Response 12:

This comment is considered identical to the next comment and will be done together in response to the next comment.

Point 13:

L.147-150 if these data were not normal distribution. Median and IQR should be expressed instead of mean (SD)

Response 13:

All data were normally distributed in the Shapiro-Wilk test, so mean (SD) was used.

Point 14:

L.151-152 please provide the unit

Response 14:

The unit "cm" has been added to L.154-155.

Point 15:

Discussion

L.172 Is there less load on the muscle also happens for MG m during land-based walking?

Response 15:

In #34, the %MVC of the quadriceps is the outcome and is not measured for MHGM. To our knowledge, no studies have reported %MVC of MHGM during gait, and the loading on MHGM is unknown.

Point 16:

Please provide more the factors that affect the changes in muscle hardness.

Response 16:

Other than what is described in L.168-173, there may be a decrease in muscle blood flow due to muscle fatigue. The paper is listed below.

The paper is listed below.

Constantin-Teodosiu, D.; Constantin, D.; Molecular Mechanisms of Muscle Fatigue. Int J Mol Sci 2021, 22(21), 11587.

Point 17:

Conclusion

Please consider to use "a decrease extracellular fluid" rather than "edema reduction". or please explain why leg edema increases during walking in that condition.

Response 17:

As you indicated, the description has been changed to "a decrease extracellular fluid", L.213-214.
